# Autonomous extraction of millimeter-scale deformation in InSAR time series using deep learning

Bertrand Rouet-Leduc [1✉], Romain Jolivet [2,3], Manon Dalaison [2], Paul A. Johnson [1] & Claudia Hulbert [2]

Systematically characterizing slip behaviours on active faults is key to unraveling the physics of tectonic faulting and the interplay between slow and fast earthquakes. Interferometric Synthetic Aperture Radar (InSAR), by enabling measurement of ground deformation at a global scale every few days, may hold the key to those interactions. However, atmospheric propagation delays often exceed ground deformation of interest despite state-of-the art processing, and thus InSAR analysis requires expert interpretation and a priori knowledge of fault systems, precluding global investigations of deformation dynamics. Here, we show that a deep auto-encoder architecture tailored to untangle ground deformation from noise in InSAR time series autonomously extracts deformation signals, without prior knowledge of a fault's location or slip behaviour. Applied to InSAR data over the North Anatolian Fault, our method reaches 2 mm detection, revealing a slow earthquake twice as extensive as previously recognized. We further explore the generalization of our approach to inflation/deflation-induced deformation, applying the same methodology to the geothermal field of Coso, California.

[1] Los Alamos National Laboratory, Geophysics Group, Los Alamos, NM, USA. [2] Laboratoire de Géologie, Département de Géosciences, École normale supérieure, PSL University, CNRS UMR 8538, Paris, France. [3] Institut Universitaire de France, 1 rue Descartes, 75005 Paris, France. ✉email: bertrandrl@lanl.gov

Faults slip in a variety of modes, from dynamic earthquakes to transient slow-slip events and aseismic slip[1,2]. The classical picture of faults being either locked and prone to dynamic and damaging earthquakes or unlocked and quietly slipping to accommodate tectonic stress is evolving. Growing evidence indicates complex fault behaviors and interactions among and between modes of slip[3]. Evidence includes fault segments hosting both slow and dynamic earthquakes, as well as slow earthquakes preceding and possibly triggering the nucleation phase of dynamic earthquakes[4–6]. Answering a number of fundamental questions, such as what controls the slip mode on a fault, whether there exists a continuous spectrum of slip modes on faults, and what determines the possible evolution of a slow earthquake into a dynamic seismic rupture, requires exhaustive characterization of all slip phenomena. Interferometric Synthetic Aperture Radar (InSAR) holds the promise of continuous geodetic monitoring of fault systems at a global scale, which may well hold the key to address these questions. However, although the data exists, current algorithms are not suited for global monitoring because they require time-consuming manual intervention, and the final product requires exhaustive expert interpretation.

InSAR is routinely used to measure ground deformation due to hydrologic, volcanic, and tectonic processes[7–9]. The apparent range change in the satellite Line Of Sight (LOS) between two SAR acquisitions is, after corrections from orbital configurations and topography, the combination of atmospheric propagation delay, changes in soil moisture and vegetation, and actual ground deformation. Rapid, large-amplitude deformation signals such as coseismic displacement fields often exceed the amplitude of sources of noise[10]. Similarly, slow but steady accumulation of deformation over long periods of time may be quantified using InSAR either through stacking[11] or time-series analysis[12,13]. However, detecting low-amplitude deformation related to transient sources such as slow-slip events, episodes of volcanic activity, or hydrologic-related motion remains challenging and requires significant human intervention and interpretation[8,14,15]. Measuring Earth surface deformation is fundamental to characterizing diverse tectonic processes, as well as surface and underground changes induced by human activities.

The most pressing issue in InSAR processing for small, millimeter-scale, deformation monitoring remains the separation between atmospheric propagation delays and ground deformation. Spatial and temporal variations in atmospheric pressure, temperature, and relative humidity modify the refraction index of the air, resulting in spatial and temporal delay variations in the two-way travel time of the radar carrier between a SAR imaging satellite and the ground[16,17]. Such delays directly affect the phase of an interferogram, which combines two SAR acquisitions. Atmospheric propagation delays in a single interferogram can be equivalent to tens of centimeters in apparent range change[16]. Current correction methods based either on empirical estimations[18,19] or on independent data[20–23] reduce the contribution of the stratified atmosphere—the long-wavelength atmospheric perturbation that, to a first order, correlates with topography. Nonetheless, remaining delays, corresponding to the turbulent portion of the troposphere may represent centimeters of apparent range change. Propagation delays in the atmosphere decorrelate after periods of 6–24 h, as shown by the temporal structure–function of Global Navigation Satellite System (GNSS) zenith delays[24]. Therefore, remaining tropospheric delays, which are coherent in space, can be considered random in the time given the time span between SAR acquisitions (e.g., 6 days for Sentinel 1, 46 days for ALOS-2). Moreover, it can be shown that, because of potential temporal aliasing[17] and loss of spatial coherence of the radar phase echo, spatio-temporal filtering can lead to biased results. Therefore, deciphering a consistent, days-to month-long transient signal in the time series of InSAR data remains a critical challenge, especially when automation is envisioned.

Convolutional neural networks are central to the most recent dramatic advances in computer vision and natural language processing. Autoencoders have been developed to create sparse representations of data—the model copies its input to its output through a bottleneck that forces a reduction of dimension equivalent to a compressed knowledge representation of the original input, enabling noise removal. Of note are recent developments applied to classify InSAR data in order to detect ground uplift and subsidence, and specifically to identify volcanic unrest[25–28]. Although promising, these developments do not make use of the different temporal signatures of signals of interest to reconstruct denoised deformation patterns.

Here, we describe a deep learning-based method to automatically detect and extract transient ground deformation signals from noisy InSAR time series. Our approach, based on a purely convolutional autoencoder, is specifically designed for removing noise in InSAR time series. In the following, we consider the evolution of the interferometric phase with time with respect to a reference both in space and time. We consider classical Small Baseline (SBAS)-like approaches for the reconstruction of the time series[29,30]. The time series we analyze stem from the inversion of a sequence of SAR interferograms previously corrected from orbital and topographic contributions[31], with a first-order atmospheric correction derived from global atmospheric reanalysis products[21,32]. Our autoencoder takes as input a noisy InSAR time series reconstructed from successive SAR acquisitions, and outputs accumulated ground deformation taking place during the time-series interval, with the atmospheric noise removed.

In this work, we first introduce the notion of autoencoders before describing the architecture of our neural network. We then describe our training set and perform preliminary tests on synthetic data. We finally highlight the efficiency of our algorithm on two reconstructed time series of ground deformation, the first one derived from COSMO-SkyMed acquisitions and the second one derived from Sentinel 1A–B SAR acquisitions.

## Results

### Description and validation of the deep autoencoder

*Autoencoder architecture.* Our goal is to extract ground deformation from noisy InSAR time series. For the purpose of training our deep learning model, we assume that input time series are the combination of three physical contributions: ground deformation, the stratified component of the atmosphere, and the turbulent component of the atmosphere. In most cases, the stratified component can be corrected for using Global Atmospheric Models (hereafter referred to as GAMs, often corresponding to reanalysis products), e.g.[32,33], or GNSS data, e.g.[34], for instance. However, such a correction is often incomplete and part of the remaining, often turbulent, atmospheric delays may still correlate with topography. Attempts have been made to estimate tropospheric delays using multispectral radiometric data[20]; however, the acquisition of such independent data must be coincident with the SAR acquisition and over a terrain with minimal cloud cover for optimal performance, conditions rarely met. In addition, it can be shown that GAM-derived correction sometimes worsens the situation as the local estimate of the state of atmospheric variables may be incorrect[32].

Our deep learning model must recognize transient deformation in InSAR time series in the presence of remaining atmospheric noise. To this end, it must distinguish the spatial and temporal

statistical differences between deformation signals and atmospheric patterns. As mentioned above, the structure of atmospheric delays decorrelates for periods longer than 6 h[24]. Ground deformation related to transient tectonic events takes place over seconds to minutes for dynamic rupture and to weeks or months or even years for slow-slip events[14,15,35,36], and remains until further deformation occurs. Therefore, the temporal signature of deformation signals is very different from that of atmospheric delays. We make use of this different temporal signature to learn appropriate filters to remove atmospheric perturbations and extract ground deformation in the InSAR time series.

Here, we build and train an autoencoding architecture to directly output the deformation signal, formulating the problem as a regression task. We rely on the following assumptions: (1) atmospheric delays are random in time, considering two successive SAR acquisitions, (2) ground deformation has a temporal coherence considering the rate at which SAR images are acquired, and (3) part of the atmospheric delay correlates with topography. We, therefore, use as inputs a time series of interferometric phase change and a map of ground elevation to produce a time series of cumulative surface displacements.

In order to separate deformation from atmospheric delays, we developed the deep learning architecture shown in Fig. 1. This architecture consists of 11 purely convolutional layers. The first six layers of the model are tasked with encoding signals that are persistent in time, by progressively removing the time dimension of the input. At the seventh layer, ground elevation (topography) is added as a secondary input. The remaining layers decode the ground deformation map. In short, we build a model tasked with reconstructing ground deformation given input InSAR time series and ground elevation from noisy input.

Initially developed for feature extraction by projecting high-dimensional data sets onto a lower-dimension manifold by forcing the reconstruction of the data through a bottleneck in deep learning architectures[37], autoencoders have also evolved into powerful denoising[38,39] and image enhancing techniques[40,41]. In this work, we exploit this aspect of deep learning autoencoding and tailor it to the problem of cleaning InSAR time series, building a deep learning autoencoder to effectively automate the design of filters in time and space to recover ground deformation.

*Training on synthetic data.* Because deep learning models require large amounts of data and there exists no ground truth for InSAR time series, we rely on synthetic data to train the deep autoencoder. The synthetic data are randomly generated cumulative surface deformation time series mimicking nine successive maps of range change. These cumulative deformation maps include surface displacements in the LOS due either to a slipping fault (either strike-slip or dip-slip) with random latitude and longitude (position in a virtual box), depth, strike angle, dip angle, and width (based on Okada's model[42]) or to an inflating or deflating point source (Mogi's model[43]). Deformation onset occurs at a random time as a pulse with a random duration within the time series, excluding the first and last time steps, which are taken as nondeforming references by the model (see Supplementary Fig. S1). The model is therefore tasked with finding cumulative deformation in the seven middle time steps of the time series arising from a wide variety of transient processes. We then corrupt each map of these ground deformation time series with different noise signals. At each time step, we create both turbulent and stratified synthetic atmospheric delays. Spatially correlated Gaussian noise mimics delays from atmospheric turbulence of various length scales[44,45] (Fig. 1, top row) and a quadratic function of pixels' elevation mimics the atmospheric delays that correlate with topography[16,46] (also randomly generated[47]). Lastly, we add random pixels, in patches and isolated, to mimic

incoherent pixels and unwrapping errors commonly encountered in real data. Each of the steps of the time-series results from a random realization of noise built following these assumptions.

We train two independent models with the synthetic time series of deformation, one tasked with recovering point source deformation and the other with recovering deformation on faults. All other phase delays are identified as noise. Both models are trained to map synthetic noisy time series to the synthetic cumulative displacements. We trained our deep autoencoder with $2.5 \times 10^7$ randomly generated time series for which we provide as input the apparent LOS deformation time series, corrupted by the sum of synthetic noise described above. The training includes a LOS with random orientation (30–45° incidence and any azimuth), so that the model is directly trained for various SAR satellite configurations and for any fault azimuth. The output is the target ground deformation accumulated during the time series. All 482,185 trainable parameters are adjusted during that training phase with the Adam variation of stochastic gradient descent[48] (see Supplementary Fig. S2 for the training curves).

We note that our deep autoencoder only considers time series of nine time steps, as a good compromise on the input duration, such that the input time series are long enough for the model to learn the temporal differences between signal and noise. When applying our models to longer time series of $n$ time steps, we apply the algorithm using a sliding window with a width of nine time steps and obtain $n - 8$ images of cumulative deformation. In this way, our model acts as a moving integral of actual deformation.

*Performance on a synthetic data set.* Once trained, we test the deep autoencoder on synthetic realizations of time series that have not been used to train the model. We randomly generate $10^5$ time series of nine time frames using the same procedure as that described for the training phase. For each of the $10^5$ time series, we evaluate the signal-to-noise ratio (hereafter referred to as SNR) as the ratio of signal power to noise power. We then apply the deep autoencoder to these time series in order to evaluate the performance of the model. We evaluate the resulting, cleaned time series using the structural similarity index[49] (SSIM, see "Methods"), a standard denoising evaluation metric, which makes a nonlocal comparison between two images, and is bound between −1 and 1.

We find that the deep autoencoder applied to synthetic data accurately reconstructs deformation signals on faults, even in circumstances very challenging to expert interpretation (SNRs well below 1; Fig. 2). For SNRs above 20%, our algorithm provides a very accurate reconstruction, as shown by the SSIM between model output and deformation ground truth (0.7 < SSIM < 1.0). For low SNRs (10% and below), no signal can be visually observed, while the structural similarity is still correct and the overall deformation signal is recovered down to SNRs of ~0.5%, below which our model starts to fail. Supplementary Fig. S1 shows the pairwise distributions of different properties of the synthetic data as well as the SNR and performance of our model. For comparison, the performance on synthetic data of the same architecture as in Fig. 1 but trained on single time steps is shown in Supplementary Fig. S3, and the performance of a simple temporal filter is shown for comparison in the Supplementary Fig. S4. The model is trained on single patches, but interestingly performs almost as well on synthetic time series with more complex fault geometry (see Supplementary Fig. S5). We note that for point sources of deformation, the limit of our model is ~20% SNR (Supplementary Figs. S9 and S10), but that such signals are also much harder to distinguish from the noise for the eye. Therefore, our architecture allows us to exceed the ability of the expert eye to detect signals in noisy time series of

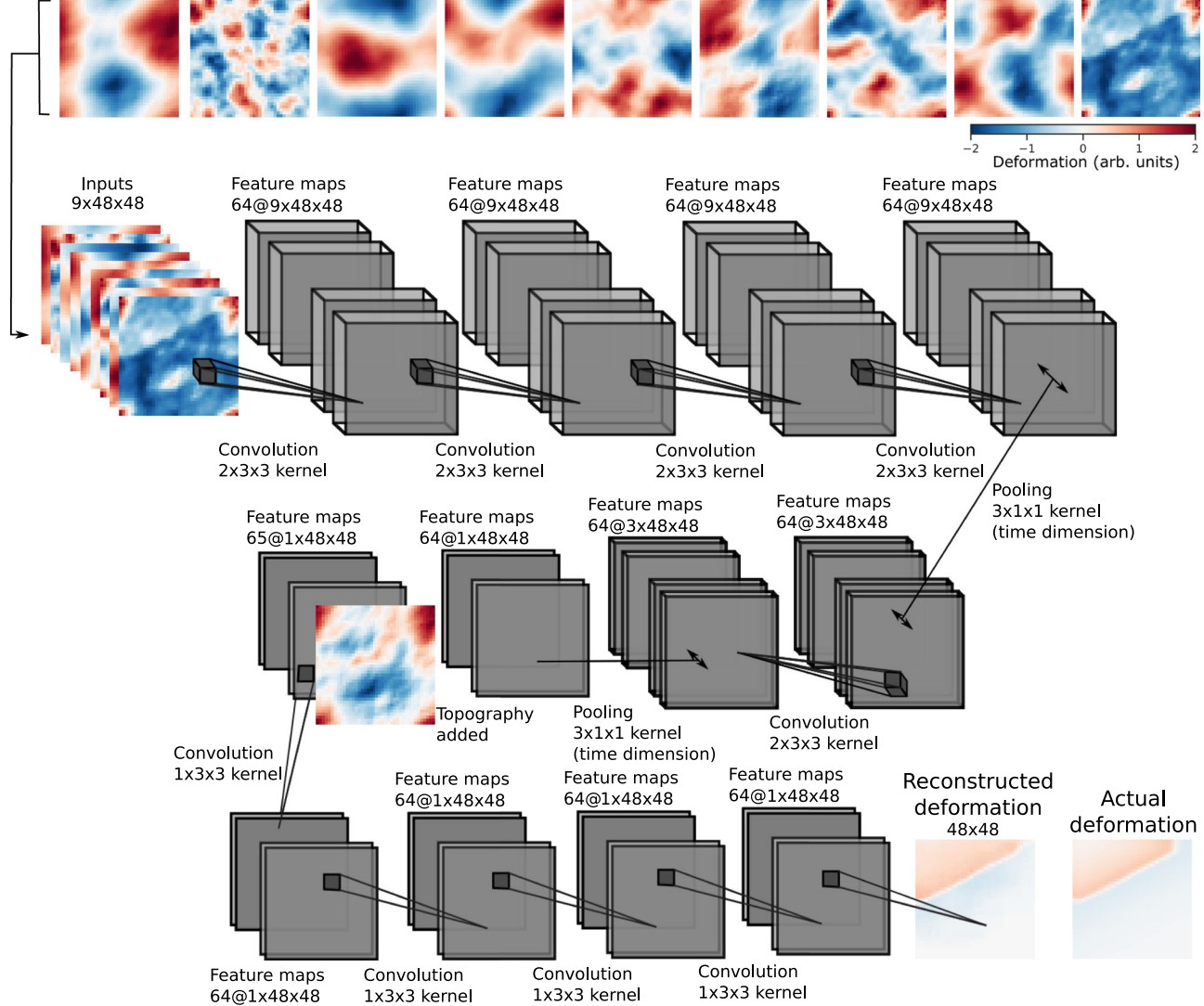

**Fig. 1 Autoencoding InSAR time series.** Schematic of our deep learning model. Top row (left to right): a sequence of synthetic InSAR time series on which the model is trained, where ground deformation signal is corrupted with atmospheric noise, including turbulence and layering of the atmosphere. Second to fourth rows: the architecture of our model. Our model is purely convolutional with progressive pooling on the time dimension during the encoding. After the time is removed, at the seventh layer, ground elevation is added as a secondary input. Fourth row: the last layers of the model are tasked with decoding ground deformation accumulated during the input time series, here compared with actual deformation that takes place in the synthetic time series shown above. A detailed description of this neural network can be found in the "Methods" section.

deformation, provided their noise structure resembles the training set.

In the following, we show the application of our autoencoder to two case studies that have been independently analyzed by InSAR experts.

### Application to real data

*Extracting deformation from a slow earthquake along the North Anatolian Fault, Turkey.* Our deep autoencoder is trained to isolate and reconstruct cumulative ground deformation signals in $48 \times 48$ pixels series of nine time steps. However, a fundamental property of purely convolutional deep learning models is that the filters they learn do not depend on input size. As a result, we can create an autoencoder with exactly the same architecture as the model described in Fig. 1, but with an input size matching the number of pixels in the InSAR time series of interest. Because the parameters of the model do not depend on the input size, we can copy every parameter (i.e., weights and biases of the filters) of the

model trained on synthetics to the new model, which can then be applied to InSAR data of any size.

Here, we apply the model to a time series built from images acquired by the COSMO-SkyMed constellation over the central section of the North Anatolian fault in Turkey (Fig. 3). This major plate boundary fault accommodates the motion of rotation of the Anatolia plate with respect to Eurasia and has ruptured in large, moment magnitude (Mw) 7 earthquakes multiple times over the past century[50]. An 80-km-long section of the fault has been slipping aseismically, at least since 1944, Mw 7.3, earthquake located near the small town of Ismetpasa[51]. In situ measurements based on creepmeters indicate that this fault experiences transient aseismic slip episodes[52–54].

Rousset et al. produced an ~1-year-long time series from COSMO-SkyMed SAR acquisitions and detected a significant slow-slip episode that lasted 1 month during 2013 with a maximum of 2 cm of fault-parallel slip[14]. Average long-term velocity maps covering the whole region derived from InSAR data show aseismic slip over an 80-km-long section of the fault. This

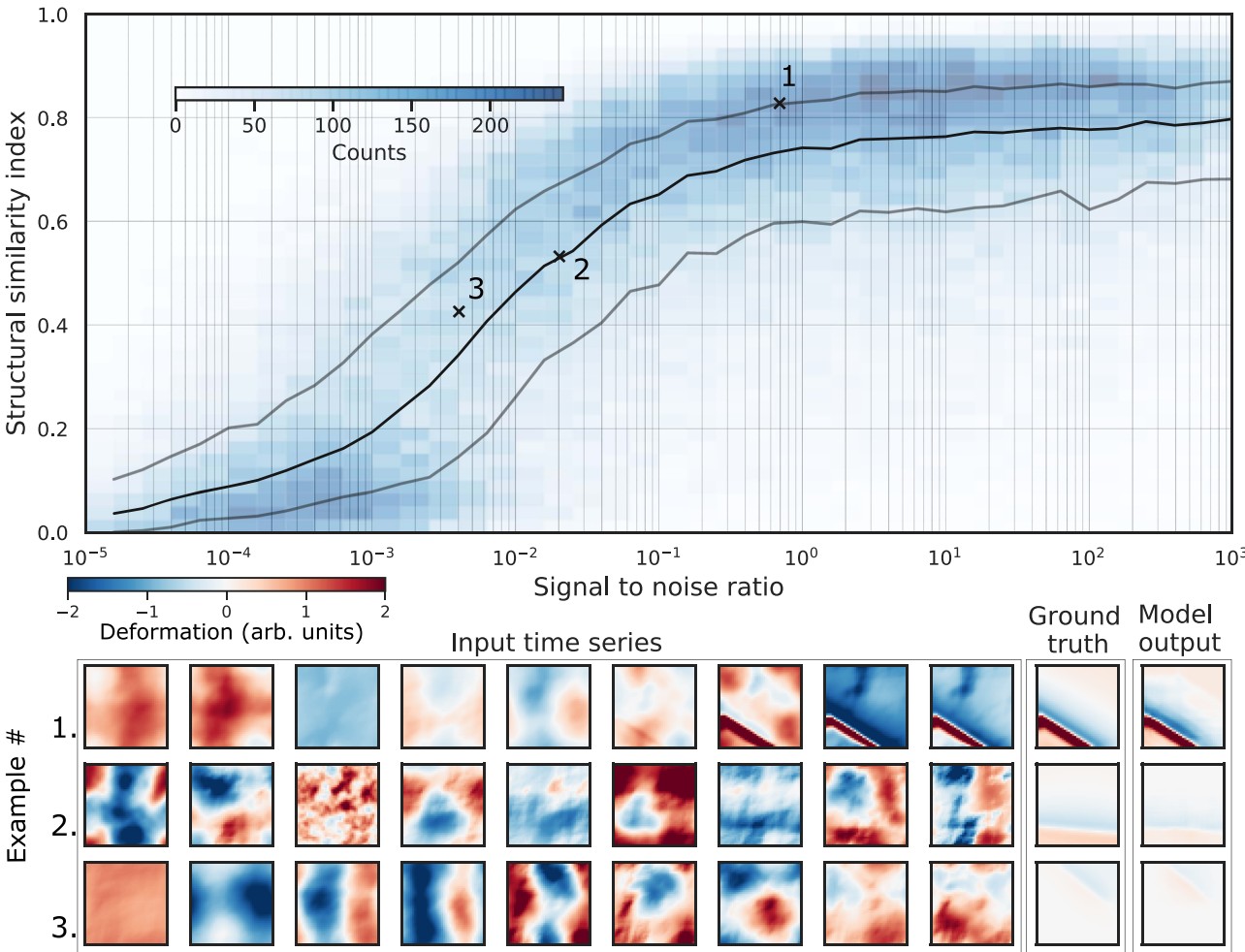

**Fig. 2 Performance on synthetic test data.** Top: performance of the reconstruction of fault deformation by our deep autoencoder, on synthetic noisy time series, as measured by structural similarity index (SSIM) between model output and deformation ground truth, as a function of signal-to-noise ratio (SNR, see "Methods"). Shades of blue show the distribution of SSIM as a function of SNR (counts per bins for $10^5$ test samples). The black and gray lines show the median and 25th and 75th percentile of the SSIM in SNR bins, respectively. Bottom: examples of the data showing input time series, ground truth, and its reconstruction, for different signal-to-noise ratios, shown with matching numbers in the plot above. Note that the model outperforms the eye, recovering with reasonable fidelity deformation signals with SNRs down to a few percent.

average relative displacement was found to result from successive transient events[14,53], which were not apparent in data from older constellations of SAR satellites due to the long time span between acquisitions. In the InSAR time series processed by Rousset et al., large atmospheric delays are apparent, despite careful correction of atmospheric delays using ECMWF reanalysis products[14,21]. Therefore, knowledge of the fault location was key in the interpretation of the surface displacement field. We revisit the same time series in order to assess if our model is able to recover the known surface slip in real-time series of data. We stress that we do not provide the location of the fault to the model. With no human intervention and no a priori knowledge of the local tectonics and fault location, the model automatically isolates and recovers clean deformation signals where expert analysis previously found signals attributed to tectonic activity (Fig. 3). Importantly, the recovered deformation is obtained after training only on synthetic data and with no further fine-tuning on real data. Our model finds up to 1.5 cm LOS relative displacement across the fault, which we interpret as the signature of surface slip, as previously found[14].

Fault-perpendicular cross-sections illustrate that even in regions where a slip would not have been convincingly identified by an expert (Fig. 4), our model recovers 2 mm of slip, extending

the previous estimate of the along-strike length of this slip event. Rousset et al. identified a 5-km-long slow-slip event while the deep learning model determines that the portion that slipped was 8.5 km in length. Interestingly, the new 2 mm slow slip we find is on a segment adjacent to the previously identified 1 cm slow slip, and the two segments are separated by a kink on the fault, suggesting a potential interplay between fault geometry and slip[55,56]. What we presume to be the remaining noise can be seen to the north-west and to the south-east of the slow-slip event in the output of the deep denoiser (see Supplementary Fig. S8 for cross-sections). We suspect these errors may arise from errors in the elevation model that propagated in the time series.

We finally note that our current model interprets wavelengths longer than a kilometer as noise, although experts might interpret those as the signature of slip at depth. This limitation however is related to the size of pixels with respect to the size of the training samples. The same network architecture trained on larger synthetics would circumvent this limitation (at the cost of increased computation and training time). An alternative approach consists of rescaling input data (see Supplementary Fig. S7) to ensure consistency of the model output in deformation wavelength (which is the case here for the North Anatolian Fault event).

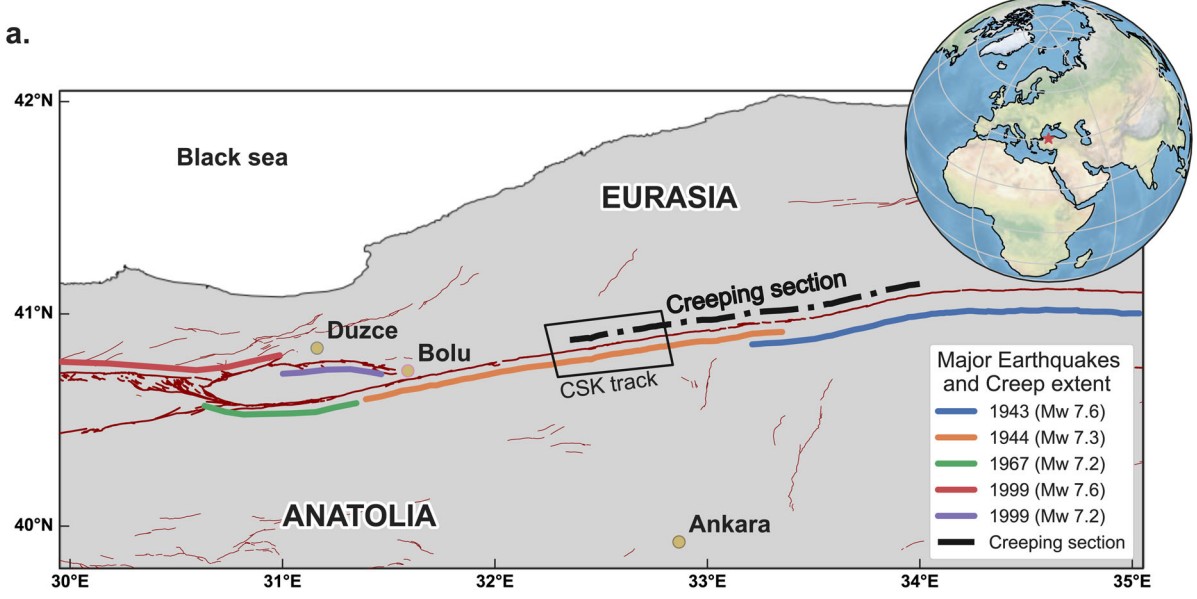

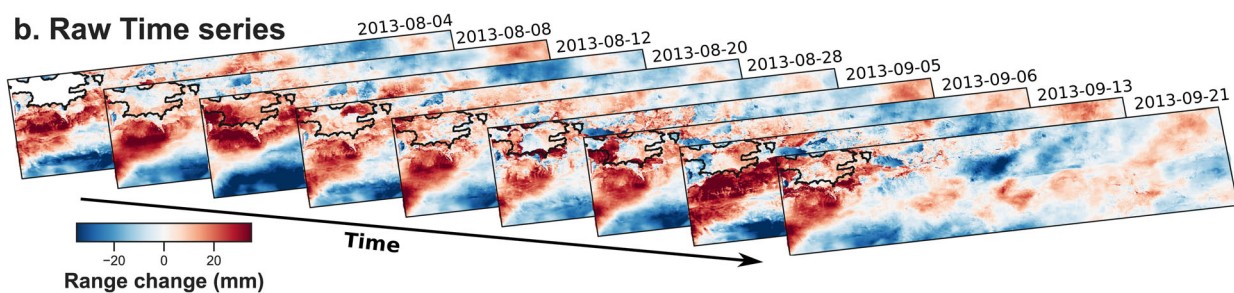

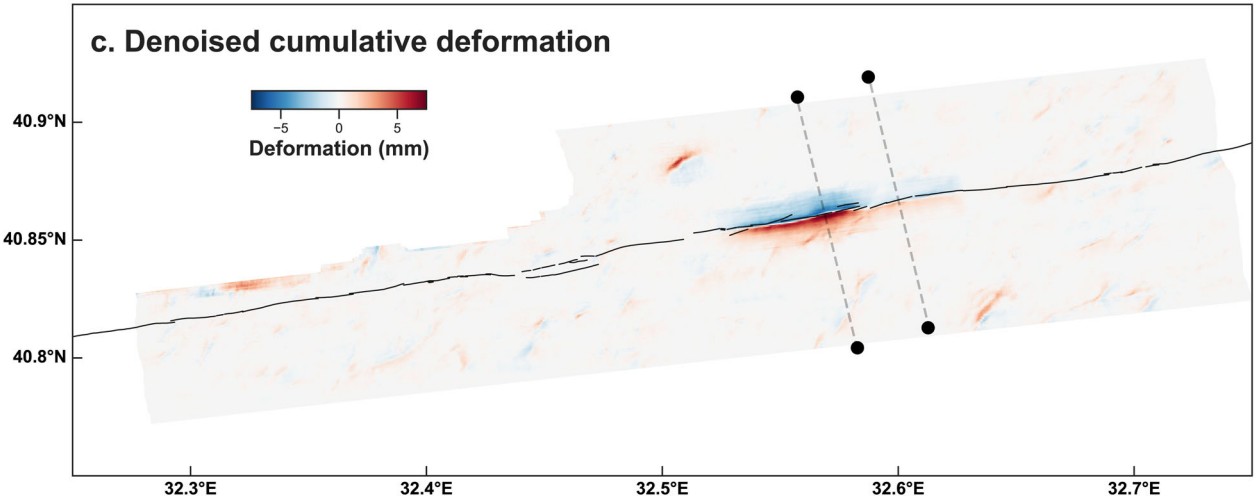

**Fig. 3 Application to real data: the North Anatolian Fault 2013 slow earthquake.** In order to identify ground deformation signals in the noisy COSMO-SkyMed InSAR time series, we create a deep autoencoder that has an input size equal to the size of each frame of the time series, 200 × 650 pixels, and the same parameters as the autoencoder trained on synthetic data, shown in Fig. 1. Inputs are the InSAR time series and the topography of the same area (not shown). The autoencoder outputs ground deformation (bottom plot). The ground deformation is manifest as an offset across the fault. The deep autoencoder finds a strong slip signal of about 1 cm (in LOS) on the fault, in agreement with previous expert analysis of the time series[14], with no a priori knowledge of the fault's existence. **a**. Seismic setting of the region of the creeping section of the North Anatolian Fault. Thick red lines are the main faults of the NAF system, separating the Eurasia plate from the Anatolia microplate. Thin red lines are other mapped structures. Colored lines indicate the extent of historical ruptures. **b** Input raw time series from COSMO-SkyMed data (a subset of the data from Rousset et al., 2016). Color is the apparent range change between the ground and the satellite. **c** Denoised cumulative deformation as output by the deep autoencoder. The color scale shows ground deformation in the direction of the LOS. Dark lines are the surface trace of the NAF, shown here for reference. Thin dashed lines indicate the cross-sections shown in Fig. 4.

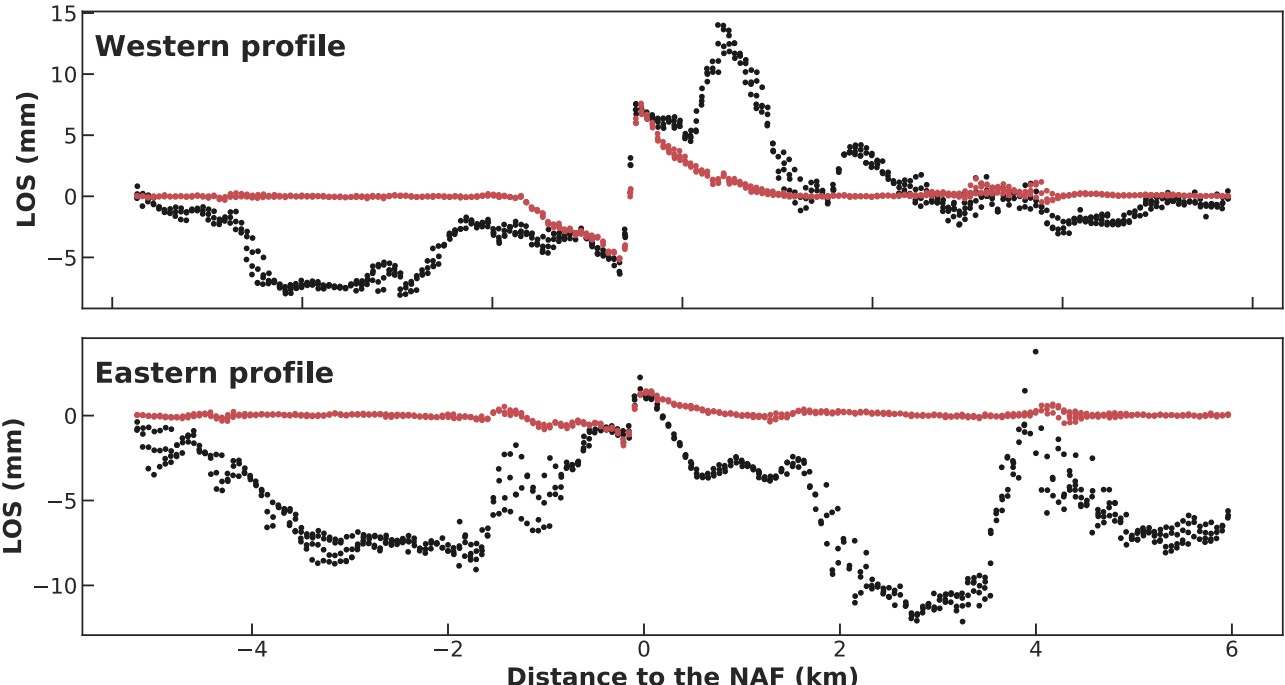

**Fig. 4 Application to real data: the North Anatolian Fault 2013 slow earthquake.** LOS deformation along fault-perpendicular cross-sections. Locations of the cross-sections are shown in Fig. 3. Blacks dots are the difference between the range change averaged between frames of the time series from 5th to 21st September 2013 (the last four frames) and the range change averaged between frames of the time series from 4th to 28th August 2013 (the first five frames), taken along a fault-perpendicular line. The main slow-slip event detected by Rousset et al. occurred during this period. Red dots are the output of the deep autoencoder highlighting the cleaned deformation pattern. The sharp offset in the input InSAR data observed exactly on the fault was interpreted as a slow-slip event by Rousset et al., in spite of the very high noise level presumably caused by atmospheric delays. Such interpretation was only made possible owing to the knowledge of the location of the fault and knowledge that this segment of the North Anatolian Fault slips aseismically. Our model knows neither and automatically extracts actual ground deformation.

*Extracting ground deformation signal at the Coso geothermal system, California*. In a second example, we use our deep learning architecture to detect surface deformation caused by underground pressure changes. As above, our model is trained on several million examples of synthetic noisy InSAR time series. In this case, surface deformation is modeled by a point pressure source using Mogi's equation of elastic deformation[57], corrupted as before by synthetic atmospheric delays. Mogi pressure sources are used extensively for the modeling of volcanic inflation and deflation signals, e.g. refs. [58,59]. Further, the combination of multiple sources allows one to model complex subsidence/uplift patterns.

After training exclusively on synthetic data, we apply our model to real data from the Coso geothermal field (California, USA), again without further training (details on the InSAR processing are in the "Methods" section). Because InSAR time series may be very noisy, even after correcting predicted atmospheric effects[32], analysis of inflation or subsidence of less than a few centimeters per year in InSAR have relied to date on deriving long-term cumulative deformation[60], such that random atmospheric delays cancel out. Detecting transient subsidence and uplift signals in SBAS time series below a few centimeters remains challenging, just as it does for faulting.

As with identifying deformation on faults, our model is able to disentangle actual ground deformation from atmospheric noise at short time scales, with a resolution of a few millimeters. In Fig. 5 we show the application of our deep denoising model to a time series over Coso in 2016. Contrary to what could be inferred from long-term cumulative deformation, we find that ground subsidence at Coso is primarily due to transient episodes of deformation. The cumulative deformation from these transients

we detect accounts for most of the cumulative deformation observed in the data (see Supplementary Figs. S11–S14 for details and for other examples of transient deformation). Interestingly, we find a number of transient events that are constituted of well-separated deflation sources, in agreement with geochemical observations showing that the geothermal field is constituted of isolated reservoirs[61].

**Discussion**

As the properties of the atmosphere cannot be measured at the same spatial and temporal resolution as SAR acquisitions, InSAR time series still contain large-amplitude atmospheric delays, on the order of centimeters, in spite of recent marked improvements in atmospheric correction and processing strategies[23,32]. For this reason, expert processing and analysis is required to interpret InSAR data. Furthermore, since the onset of the Sentinel 1 mission, the amount of available InSAR data has grown at a pace that is already challenging the ability of the community to process and analyze it, and the upcoming NISAR mission will increase the amount of available InSAR data several fold. Therefore, significant effort has been put into developing strategies to build time series with such vast data sets, e.g., refs. [13,30,62]. Nonetheless automatic, autonomous InSAR interpretation methods are poised to become essential, if just to leverage the increasing spatial and temporal resolution of the data.

We note that several avenues of improvement should enhance the ability of our neural network to detect finer and finer deformation signals in the future. First, we did not include sources of noise representative of ionospheric perturbations. The total electronic content of the ionosphere introduces a differential

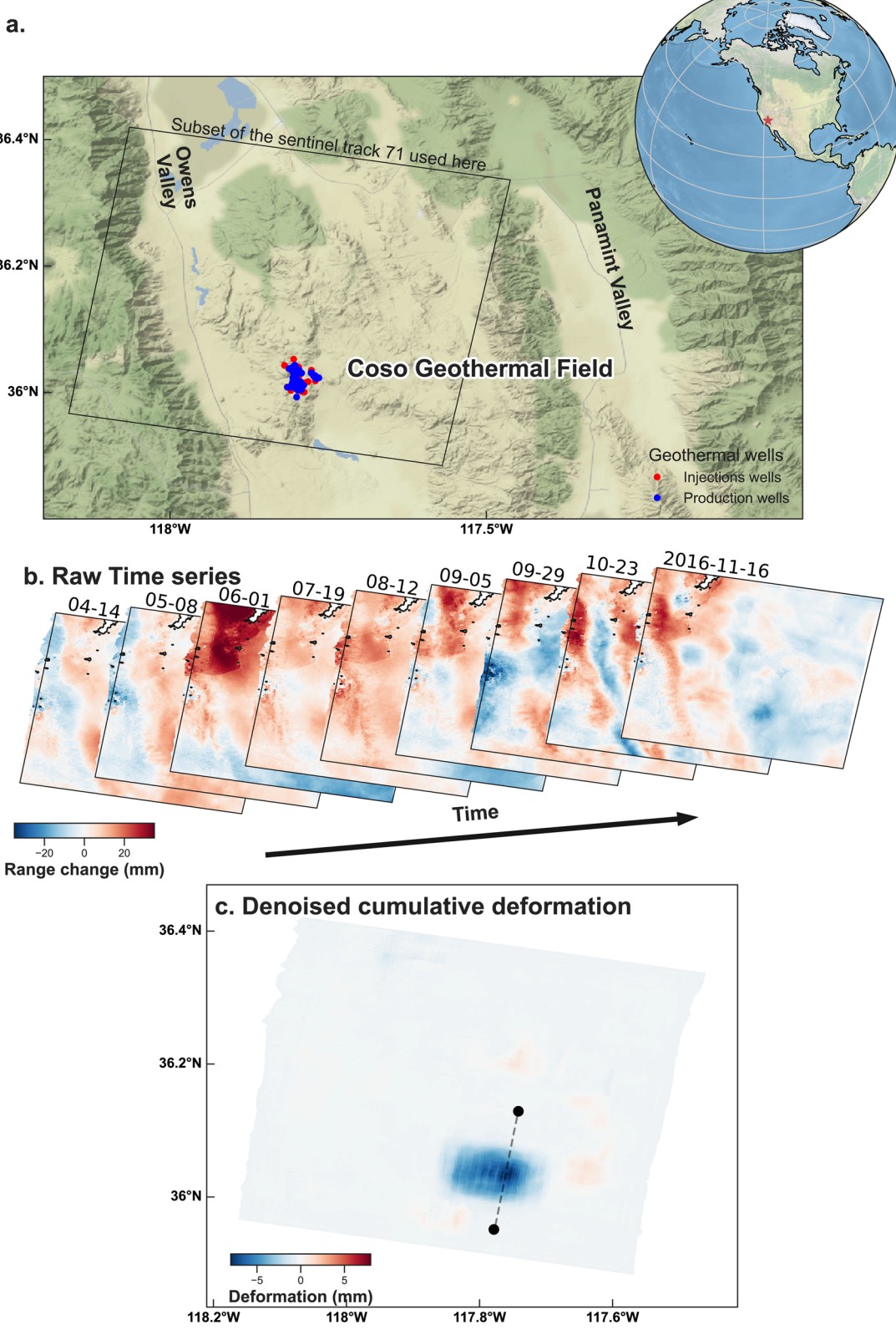

delay in interferograms that can bias analysis further[16]. Although this effect is more pronounced for L-band SAR satellites[16,63], long-wavelength ionosphere delays can be problematic for large images acquired with C-band SAR systems such as Sentinel 1[64]. Although these delays can be corrected by using techniques such as the range split-spectrum method[64,65], the structure of the remaining noise associated with imperfect corrections must still be evaluated and could then be used in the training of our model.

Second, we considered atmospheric turbulence to be isotropic and equivalent everywhere in the image (i.e., noise is second-order stationary) while some anisotropy can be observed in the phase delay of some interferograms. However, such anisotropy depends on the scale of the image observed, which would involve complex considerations in the construction of an adequate tropospheric noise model to train our model. In general, any improvement in the forward modeling of the nature of noise in

**Fig. 5 Application to real data: the Coso Geothermal Field in California.** After training our deep autoencoder architecture exclusively on synthetic InSAR time series of point sources of deformation corrupted with atmospheric noise, we apply it to the time series obtained from Sentinel 1A–B from 14 April 2016 to 16 November 2016, which spans the Coso Geothermal Field in California. Our model detects a transient episode of subsidence of 5–7 mm (in line of sight), where the operational wells are located, with no a priori knowledge of the area. **a** Geographic setting with the coverage of the subset of the Sentinel 1 track used here. Red and blue dots indicate the geothermal wells, respectively, for injection and production. **b** Input raw time series of nine successive images from Sentinel 1 data. Color is the apparent range change between the satellite and the ground along the LOS. **c** Denoised cumulative deformation as output by our deep autoencoder. Color is ground deformation in the LOS. The thin dashed line indicates the location of the cross-section shown in the Supplementary.

---

InSAR should lead to a significant improvement in the detection capability of the models. Finally, the receptive field of the auto-encoder and the pixel size of the input InSAR data restrict the size of the deformation signal that can be deciphered. For instance, interseismic deformation related to loading of a fault by plate motion extends over 10 s of kilometers, e.g., refs. [11,12,66]. Additional developments may be necessary for the detection and cleaning of long-wavelength deformation patterns.

The initial application of our method on InSAR time series enables the direct observation of a slow earthquake, refining previous estimates, autonomously and without prior knowledge. In particular, we expect that the ability to systematically observe fault and pressure source deformation at a global scale will further the understanding of hydrologic, volcanic, and tectonic processes, and may bring us closer to bridging the observational gap that exists for transient surface deformation.

## Methods

**Autoencoder architecture.** Here, we provide additional details regarding the autoencoder architecture developed for ground deformation extraction from InSAR time series (Fig. 1). This architecture consists of 11 purely convolutional layers. The first six layers of the model are tasked with encoding signals that are persistent in time, by progressively removing the time dimension of the input. At the seventh layer, topography (a digital elevation model) is added as a secondary input, before the remaining layers decode the ground deformation map. Because our model is comprised of purely convolutional layers, it can be applied to arbitrarily sized inputs (in terms of the spatial dimension, not the time dimension because of the pooling in time).

At each layer, the input is passed through 64 different filters to form as many channels, which are simplified representations of the data. During the encoding, 3D filters (two dimensions of space and one in time) of size $3 \times 3 \times 2$ are applied to the data, until time has been completely eliminated by max-pooling operations. During decoding, 2D filters of size $3 \times 3$ are applied to the data (two dimensions of space) and summed for the output layer to reconstruct the cumulative ground deformation.

For each of the 64 filters within the encoder layers, each of the input filtered channels are summed and passed through a biased leaky reLU activation function. Each layer has a number of trainable parameters given by $n_{kernel} \times n_{input} \times n_{output} + n_{output}$, with $n_{kernel}$ the convolutional kernel size (product of its shape in all dimensions), $n_{input}$ the number of input channels to the layer, and $n_{output}$ the number of output channels of the layer. This procedure gives $3 \times 3 \times 2 \times 64 \times 64 + 64$ trainable parameters for each encoding layer, except for the first one, which has $3 \times 3 \times 2 \times 1 \times 64 + 64$ parameters, and $3 \times 3 \times 64 \times 64 + 64$ trainable parameters for each decoding layer, except for the layer where ground elevation is added as an additional channel, that has $3 \times 3 \times 65 \times 64 + 64$ trainable parameters, and except for the last decoding layer that has $3 \times 3 \times 64 \times 1 + 1$ trainable weights. This gives our deep autoencoder a total of 482,185 trainable parameters, a modest amount when compared to natural image classification networks such as AlexNet[67], which has 62,378,344 trainable parameters. The last layer of our model has a linear activation instead of a leaky reLU, such that positive and negative deformations can be equally output for the final reconstruction. Final reconstruction is a single image of the cumulative deformation that occurred during the nine time steps used as input. Our model was implemented on GPUs using the keras and tensorflow python libraries.

**Evaluation metrics.** To assess the performance of our model on synthetic test sets, we use the SSIM, a common denoising performance metric in image processing. This measure of resemblance between two images is nonlocal and compares intensity, luminance, and contrast of the two images in moving windows, resulting in a metric closer to perceived similarity. We use the formulation and parameters

from the original paper[49]:

$$\text{MSSIM}(\mathbf{X}, \mathbf{Y}) = \frac{1}{M} \sum_{j=1}^{M} \text{SSIM}(\mathbf{x}_j, \mathbf{y}_j) \tag{1}$$

$$\text{SSIM}(\mathbf{x}, \mathbf{y}) = \frac{(2\mu_x \mu_y + C_1)(2\sigma_{xy} + C_2)}{(\mu_x^2 + \mu_y^2 + C1)(\sigma_x^2 + \sigma_y^2 + C_2)} \tag{2}$$

$$\mu_x = \sum_{i=1}^{N} w_i x_i \tag{3}$$

$$\sigma_x = \left( \sum_{i=1}^{N} w_i (x_i - \mu_x)^2 \right)^{1/2} \tag{4}$$

$$\sigma_{xy} = \sum_{i=1}^{N} w_i (x_i - \mu_x)(y_i - \mu_y), \tag{5}$$

with $C_1 = (0.01L)^2$, $C_2 = (0.03L)^2$, $L$ the range of the pixel values, $x_i$ the pixel values of patch $\mathbf{x}$ of image $\mathbf{X}$, and $w_i$ weights given by the unit Gaussian function with a standard deviation of 1.5 pixels. The SSIM values we report here are the average SSIM of aligned patches $\mathbf{x}$ and $\mathbf{y}$ of size $8 \times 8$ from the two compared images $\mathbf{X}$ and $\mathbf{Y}$.

We use a standard definition of SNR, as the ratio of signal power to noise power:

$$\text{SNR}(\mathbf{X}) = \frac{\text{P}(\mathbf{X}_{signal})}{\text{P}(\mathbf{X}_{noise})} = \frac{\text{RMS}^2(\mathbf{X}_{signal})}{\text{RMS}^2(\mathbf{X}_{noise})} \tag{6}$$

$$\text{RMS}(\mathbf{X}) = \sqrt{\frac{1}{9} \frac{1}{N^2} \sum_{t=1}^{9} \sum_{i,j=1}^{N} x_{ijt}^2} \tag{7}$$

where $\mathbf{X}_{signal}$ is a nine frames long time series of synthetic deformation (e.g., deformation on a fault patch), and $\mathbf{X}_{noise}$ is a nine frames long time series of synthetic noise, as described in the main text.

**COSMO-SkyMed data processing over Turkey.** We use the ISCE framework to combine COSMO-SkyMed acquisitions into coregistred interferograms and then filter and unwrap these interferograms[14,68]. After interferogram generation, we use the ERA-Interim atmospheric reanalysis to perform a first-order correction of the atmospheric phase delay[21,32]. Finally, we use the New Small Baseline Subset (NSBAS) method implemented in the GIAnT toolbox[69] to construct a time series of phase change, e.g.[12] (extended details about the processing and the data set can be found in ref. [14]).

**Sentinel 1 InSAR time series over Coso.** We process the Synthetic Aperture Radar (SAR) images collected along ascending track 64 of Sentinel 1A–B from October 2015 to July 2019. We build 244 unwrapped interferograms (Supplementary Fig. S15) using the ISCE package[68]. We coregister SAR images with a network-based enhanced spectral diversity approach[70] and correct for atmospheric perturbations using ERA-5 ECMWF global reanalysis of atmospheric data[21]. We apply a phase-preserving filter and multilooking (i.e., averaging of adjacent pixels) so that the final pixel size is about 70 m in range and azimuth[71]. Potential unwrapping errors are corrected using CorPhu[72]. Interferograms are unwrapped using a branch-cut method[73] in areas for which coherence exceeds 0.5. We subtract a best-fitting ramp (i.e., linear function in range and azimuth) to each interferogram to correct long-wavelength perturbations due to orbital errors or ionospheric content, in order to focus on local, kilometer-scale deformation. Finally, we computed the optimal time series of displacement with the NSBAS approach as it is implemented in GIAnT[74]. Any pixel for which one interferogram could not be unwrapped is not included in the study. This restrains the spatial coverage of our InSAR time series, but ensures maximum (and equivalent) redundancy to all pixels.

## Data availability

All the InSAR data used here is freely available from the European Space Agency. The COSMO-SkyMED archives and the Sentinel 1 data can be found at https://earth.esa.int.

## Code availability

The synthetic data used to train the model are based on the open-source code CSI from R. Jolivet and can be found at http://www.geologie.ens.fr/jolivet/csi/. The deep learning model has been developed using the open-source Python package, Tensorflow.

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

## Acknowledgements

B.R.-L.'s work was funded by Institutional Support (LDRD) at Los Alamos (20200278ER). R.J., M.D., and C.H. were supported by the European Research Council (ERC) under the European Union's Horizon 2020 research and innovation program (Geo-4D project, grant agreement 758210). C.H. was also supported by the CEA-ENS Yves Rocard LRC (France). P.A.J. was supported by the DOE Office of Science (Geoscience Program, grant 89233218CNA000001) and LDRD. We thank Chris X. Ren for his comments on the paper.

## Author contributions

Author order uses the remote sensing convention of author contribution. B.R.-L. and R.J. formulated the problem as a deep denoising task. B.R.-L. created the deep learning model and applied it on real InSAR data, with help from R.J., M.D. and C.H.; R.J. implemented the synthetic data used for training the model and processed the COSMO-SkyMED InSAR data; M.D. processed the Sentinel 1A–B data. All the authors analyzed the results and wrote the paper.

## Competing interests

The authors declare no competing interests.
