## [Peer Review File · Nature Communications]

Autonomous Extraction of Millimeter-scale Deformation in InSAR Time Series Using Deep LearningReviewers' Comments:

Reviewer #1:

Remarks to the Author:

This paper describes an approach to denoising InSAR time series using a convolutional auto-encoder, and applies the approach to a slow slip transient on the NAF and subsidence of the the Coso geothermal system. The approach is novel and produces some really exciting results - techniques like this are set to revolutionise the way we deal with InSAR data, and this is a fantastic contribution.

I do wonder how faithfully the results represent the true deformation pattern, however. In the NAF case, the auto-encoder clearly extracts the offset across fault, but how much confidence can we have in the shape of the curve away from the fault? The profiles in Fig. 4 look like an almost perfect scaled $\pi/2$ -arctangent plot, which would be expected for a uniform slip model. But could this be only because all the training models look like this? The same question arises for the point source model. Maybe this could be tested by introducing synthetic deformation sources into time series of real data for which the auto-encoder currently finds no deformation. The synthetic deformation sources could have more complex patterns, as typically seen in real data.

I've listed some other comments below, which can hopefully be used to polish the manuscript, but I would certainly like to see this published when ready.

L36: other signals like changes in soil moisture and vegetation can also contribute.

L90: I would say the description and validation of the auto-encoder is not "results" and should have its own section.

L141: By 9 'acquisitions' do you actually mean 9 interferograms?

L143: Is fault slip modelled as a single patch? And is rake random too? All the examples look like they might be strike-slip, but it would be good to state if this is the case. If so, presumably, the auto-encoder, as trained, only works for strike-slip faults.

L144: "Deformation onset occurs at a random time with a random duration": is the deformation stationary in space and steady-state in time?

L145: Having been trained on time series with no new deformation in the 1st or 9th image, what happens if the test data do have deformation in these images?

Fig 2: Does R^2 change if deformation is visible in all seven middle images or only the last one? Perhaps you could make plots of how R^2 varies, for a fixed SNR, with various factors such as rake (if it is allowed to vary), slip duration, number of images etc.

Fig 2: "the model outperforms the eye": but the model has access to more data than the eye, which only sees the final interferogram.

Fig2: Please add a colour scale.

L184-192: Should be mentioned that a parallel figure for point sources is available in Supplementary, and that the performance is not as good.

L184-192 Is the performance as good when deformation partly correlates with topography? This is perhaps most likely in the pressure source case.

Fig. 3: There are off-fault signals to the NW and SE of the slip event, which have similar magnitude to

the eastern part of the event ($\sim 2\text{mm}$). Are these real or artifacts? How do profiles look across them? Do they also have the perfect arctangent shape?

Fig. 3: Please add the time of acquisitions in the caption.

L200: How are areas with no data treated? It looks like there may be values everywhere for the NAF example, but this won't generally be the case, and it looks like there are data gaps in the Coso case. If there were missing data in a region of subsidence, would this cause a problem?

L235: The length of the slipping portion is stated to be 15-20 km, but from Fig. 3 it looks to be ~ 10 km (about 0.1 degree).

Fig 4: I'm not sure the black dots are a fair comparison, as they are not derived from all of the 9 interferograms that the auto-encoder uses. Fitting an appropriate function in time could generate a fairer comparison.

L261: Most of the well-separated deflation sources seem to be some distance from the wells – how do you explain this?

Equation 2: This is an odd definition for SNR, which is conventionally defined as the ratio of signal power to noise power. A better calculation would be $(\text{RMS of the signal})^2 / (\text{RMS of noise})^2$

Supp. Fig. 3: How is the cumulative figure actually calculated?

Supp. Fig. 3: The following statement is made "most of the deformation at Coso takes place as transient deformation (which is what the model is sensitive to)" If the deformation was linear in time, wouldn't most of it still appear when summing up a series of outputs from the auto-encoder?

Supp. Fig. 5 (and 3): It looks like there is an area of unwrapping errors. Is this the same area that is interpreted as an area of uplift (it looks like it is actually to slightly offset)? If so, it might be having a similar effect as introducing synthetic deformation as I suggested above. Some discussion of the effect of unwrapping errors would be useful, in any case.

Minor typo:

L22 's' missing from 'slow earthquake'.

Andy Hooper

Reviewer #2:

Remarks to the Author:

InSAR has become one of the recent advances in high-resolution mapping of large-scale ground deformation on a regular basis. However, atmospheric phase delays have been a major factor that continues to hamper the accuracy of ground deformation measurements. Accurately extracting mm-scale deformation from InSAR timeseries hence is a critical yet challenging topic.

The authors propose a deep convolutional autoencoder to denoise InSAR timeseries that are contaminated by atmospheric noise, where the autoencoder learns the different spatial-temporal statistics between signal and noise, and outputs the cumulative displacement in the time series. To the best of my knowledge, this is the first study that utilizes inherent spatial-temporal statistics between signal and noise for denoising, though convolutional neural networks have been applied on individual interferograms in several previous studies. The proposed autoencoder is shown to be effective in extracting slow slip events on the NAF with 2-mm-level accuracy and ground deformation

over the Coso geothermal field.

The study is interesting and promising. Once proved to be valid, the deep-learning based autoencoder could be a powerful tool to reveal ground deformation, especially for transient processes (e.g. SSE, transient creep) that had been previously hidden by atmospheric noises. I found the paper suitable for publication in Nature Communications once the authors addressed the comments below:

Questions about the autoencoder training:

1. At the 7th layer of the autoencoder, ground elevation is added. In the training set, is the topography randomly generated, or does it use the topography of the corresponding study area? Could you please elaborate on that. That is, can the pre-trained autoencoder be generalized to any study region if characteristics of DEMs are considered during training, or does the autoencoder need to be trained on a case-by-case basis?
2. If the input topography in the training set is from the study region, how is the topography is handled when converting it to the size of 40*40? Would sharp topographic boundaries be smeared out during the process? If so, the autoencoder may not be able to learn topography-correlated noise features below a certain spatial scale.
3. For how many epochs did the training go? What is the training accuracy? Please provide the learning curve for the autoencoder (R^2 versus training epochs).
4. Is data normalization applied for input synthetic timeseries in the training set?

Questions about applications on real datasets:

5. The underlying assumptions for the implementation of the proposed autoencoder is that the signal is coherent whereas the atmospheric noises are random in time (Line 117). What if the deformation-signal duration is shorter than the time interval of the two SAR acquisitions? Would short transient signals hence be considered as noise?

For slow slip transient detection, this is probably not an issue for Sentinel and upcoming NISAR mission given their frequent recurrence interval. However, for ALOS-2, transient signals with <42 days duration may be falsely regarded as noise.

To better address this issue and evaluate the autoencoder performance, the authors could show the training accuracy versus the signal duration. A dependency may be expected. This evaluation is crucial when interpreting the cleaned timeseries outputs.

6. In Figure 3, several sub-structures are observed along with the NAF slow slip events, e.g. features at (32.5°E, 48.88°N) and (32.63°E, 48.8°N). It is exciting to see those subtle features being extracted out of the noisy dataset. But it might also be residual noise. Any comments on those second-order features? What are the expectations when interpreting outputs from the autoencoder?
7. The authors indicate that the current model interprets signals with wavelengths longer than 1 km as noise (Figure 4 caption). However, in the training set, the fault parameters are randomly set, including the fault depth. What I understand here is that the autoencoder is designed to not only identify surface shallow slip but also slip at depth. What is the range of fault depth? The maximum wavelength it could resolve would correspond to the features of the input training set. Please clarify.
8. Please indicate how the NSBAS network is formed in the Data Section, if it is not previously published. Figure S1 does not seem to match the description in Line 350. Missing figure?

Reviewer #3:

Remarks to the Author:

This paper introduces a deep auto-encoder that is designed to untangle ground deformation from atmospheric noise in InSAR time-series. I enjoyed reading the paper as it is very well written with clear and demonstrative examples.

The following are some detailed comments:

(1) I find it slightly awkward branding de-noising InSAR cumulative deformation as "autonomous" extractions of deformation. InSAR processing itself does not involve prior knowledge of local tectonics. With good SNR, even the simplest InSAR processing can manifest deformation that can be identified without expert eyes -- and we do not brand SBAS as a method to "autonomously" extract signals. The presented model is an additional step in standard InSAR processing (since it requires the input of InSAR time-series) designed to retrieve small signals in low SNR scenarios. The extracted deformation from the presented model still needs expert interpretation to avoid false positives, whether because the synthetic noise model in the training sets does not represent actual noise in the data or because the model failed with very low SNR.

(2) Given how much the performance of the model heavily depends on the design of the relative spatial wavelengths of the noise and signal in the training sets (the authors noted, first in Fig. 4 that the model interprets wavelength longer than 1 km as noise and then in the discussion that the input pixel size of the InSAR data restricts the size of the deformation signal that can be deciphered), I am interested to see how well temporal filtering worked. Given that making use of different temporal signatures of deformation and atmospheric signals is the main novel aspect in terms of the model design of the paper, I think the paper will benefit from some discussions on this aspect.

Response to Reviewers

We are very grateful for the time and effort spent by the reviewers on our paper, and we have tried to address every comment and question in the detailed response below.

In what follows the reviewers' comments are in black and our answers are in blue. The edited text in the manuscript and its supplementary is also in blue.

Reviewer #1

This paper describes an approach to denoising InSAR time series using a convolutional auto-encoder, and applies the approach to a slow slip transient on the NAF and subsidence of the the Coso geothermal system. The approach is novel and produces some really exciting results - techniques like this are set to revolutionise the way we deal with InSAR data, and this is a fantastic contribution.

I do wonder how faithfully the results represent the true deformation pattern, however. In the NAF case, the auto-encoder clearly extracts the offset across fault, but how much confidence can we have in the shape of the curve away from the fault? The profiles in Fig. 4 look like an almost perfect scaled $\pi/2$ -arctangent plot, which would be expected for a uniform slip model. But could this be only because all the training models look like this? The same question arises for the point source model. Maybe this could be tested by introducing synthetic deformation sources into time series of real data for which the auto-encoder currently finds no deformation. The synthetic deformation sources could have more complex patterns, as typically seen in real data.

I've listed some other comments below, which can hopefully be used to polish the manuscript, but I would certainly like to see this published when ready.

Dear Prof. Hooper, many thanks for your warm encouragements. We have tried to address all your questions, notably by performing a number of tests on synthetic data.

L36: other signals like changes in soil moisture and vegetation can also contribute.

Absolutely, we have changed the sentence accordingly: "The apparent range change in the satellite Line-Of-Sight (LOS) between two SAR acquisitions is, after corrections from orbital configurations and topography, the combination of atmospheric phase delay, changes in soil moisture and vegetation, and actual ground deformation" (l., 36).

L90: I would say the description and validation of the auto-encoder is not "results" and should have its own section.

We have made the corresponding change.

L141: By 9 'acquisitions' do you actually mean 9 interferograms?

We mean 9 time steps of the deformation time series built by NSBAS, for example. We have changed the wording throughout the text and in the caption of Fig. 3 and 4.

L143: Is fault slip modelled as a single patch? And is rake random too? All the examples look like they might be strike-slip, but it would be good to state if this is the case. If so, presumably, the auto-encoder, as trained, only works for strike-slip faults.

The model is trained on single patches, but interestingly performs just as well on synthetic time series with more complex fault geometry (see new supplementary figure, S4). For instance, in the case of the NAF slow slip event, the signature of surface slip is not a straight line and follows the surface trace of the NAF quite remarkably.

During the training, patches are randomly either pure strike-slip or pure dip-slip, and we added some details to that effect in the text: “due either to a slipping fault (either strike-slip or dip-slip) with random latitude and longitude” (l.142). We also added a supplementary figure that shows the performance of the model depending on parameters of the fault slip, which shows little difference for strike slip and dip slip (new figure S1).

L144: “Deformation onset occurs at a random time with a random duration”: is the deformation stationary in space and steady-state in time?

The deformation is stationary in space and pulse-like in time (i.e. it has the shape of the cumulative density function of a normal distribution). We added the precision: “Deformation onset occurs at a random time as a pulse with a random duration within the time series” (l. 145).

Interestingly, although it is trained on stationary slip it also catches propagating slip (not shown here). We presume this is for the same reason that, although the model is trained on single patches, it still works on multiple patches.

L145: Having been trained on time series with no new deformation in the 1st or 9th image, what happens if the test data do have deformation in these images?

In training there was deformation in the 1st and 9th images, but they were used as references not to be reconstructed. As a result the model is insensitive to deformation in the 1st and 9th images (see new supplementary figure S5). If deformation is occurring at these time steps, a set of 9 overlapping images before or after should catch the deformation pattern.

Fig 2: Does R^2 change if deformation is visible in all seven middle images or only the last one? Perhaps you could make plots of how R^2 varies, for a fixed SNR, with various factors such as rake (if it is allowed to vary), slip duration, number of images etc.

We made a few such tests that are summarized in a new supplementary figure S1. We performed those tests in the SNR region where the performance is flat ($\text{SNR} > 1$) and see little to no effect of rake (dip slip versus strike slip) slip duration or slip occurrence.

Fig 2: “the model outperforms the eye”: but the model has access to more data than the eye, which only sees the final interferogram.

We added the entire inputs to make a fairer comparison and maintain that the model outperforms the eye. The model still performs very well around SNRs of 5% where we are unable to see anything.

Fig2: Please add a colour scale.

We made the change in Fig. 2.

L184-192: Should be mentioned that a parallel figure for point sources is available in Supplementary, and that the performance is not as good.

We have added the following clarification: “We note that for point sources of deformation, the limit of our model is approximately 40% SNR, but that such signals are also much harder to distinguish from noise for the eye.” (l. 191).

L184-192 Is the performance as good when deformation partly correlates with topography? This is perhaps most likely in the pressure source case.

The new figure S1 shows that correlation between signal and topography has little to no effect on detecting deformation from slip. We also performed similar tests for pressure sources, and the results are in a new supplementary figure S9, that shows a slight negative dependence between model performance and correlation between deformation and topography.

Fig. 3: There are off-fault signals to the NW and SE of the slip event, which have similar magnitude to the eastern part of the event ($\sim 2\text{mm}$). Are these real or artifacts? How do profiles look across them? Do they also have the perfect arctangent shape?

We are aware that here and there, some signals are considered by the neural network and kept after the denoising step. For instance, the 2 relatively large signals pointed out by Pr. Hooper stand out in the results shown in the NAF case and could be considered as slip along faults. We do not know why these appear in the results and cannot really attribute them to any distinguishable feature on the field. Their shape is not perfectly symmetric (but still quite symmetric, indeed) and they are not located along any known structure. Therefore, we consider these as false positives left by the neural network. We added in the supplementary information a new figure that highlights these false positives. We also now mention in the main text (l. 245) that, for some reason that we cannot explain yet, false positives seem to arise with a preferred orientation, 45° with respect to the LOS. Potential errors might come from us

not accounting for DEM errors, which could translate into a signal that would correlate with the gradient of topography (although that would not explain the 45° orientation). In future work we will try to understand the source of these false positives and correct for them.

Fig. 3: Please add the time of acquisitions in the caption.

We made the change to the figure.

L200: How are areas with no data treated? It looks like there may be values everywhere for the NAF example, but this won't generally be the case, and it looks like there are data gaps in the Coso case. If there were missing data in a region of subsidence, would this cause a problem?

We have two main options for missing data: either let it propagate through the network, as is seen in Fig. 3, or replace all the missing data by random noise (uniformly distributed), which doesn't affect the output of the model. We used the second option for the Coso data.

L235: The length of the slipping portion is stated to be 15-20 km, but from Fig. 3 it looks to be ~10 km (about 0.1 degree).

Indeed, we now re-measured the length of the known portion that slipped by more than a cm, which is about 5 kms. The inversion that Rousset et al. performed also finds about 5 kms of slip rupture length (their figure 4). The newly detected portion that slipped by a couple millimeters adds about 3.5 kms of rupture. We have made the modification in the text (ll. 241 and 243).

Fig 4: I'm not sure the black dots are a fair comparison, as they are not derived from all of the 9 interferograms that the auto-encoder uses. Fitting an appropriate function in time could generate a fairer comparison.

The black dots do use the 9 time steps, they are the average range change after the slow slip minus the average range change before the slow slip. We have tried to make it clearer in the caption. Because the noise is auto-correlated in time, these averages are still very noisy.

L261: Most of the well-separated deflation sources seem to be some distance from the wells – how do you explain this?

In the figure in the main text, we show an example in which the deflation sources sit on top of the wells. We do not know why some other sources arise from areas where no wells are located in some other cases and we wanted to show these examples as potential detections of unexpected deflation/inflation signals. These remain to be investigated.

Equation 2: This is an odd definition for SNR, which is conventionally defined as the ratio of signal power to noise power. A better calculation would be $(\text{RMS of the signal})^2 / (\text{RMS of noise})^2$

We are now using this definition and changed all the figures accordingly. In the same vein of changes, we also replaced R2 (coefficient of determination) with SSIM (structural similarity index), which is a more common metric for assessing denoising methods.

Supp. Fig. 3: How is the cumulative figure actually calculated?

The cumulated figure from the outputs of the model is made using the output of the deep denoiser as deformation rate and integrating it over the entire time series. Temporally reconstructing the output of the deep denoiser is still a work in progress. The cumulated figure for the InSAR data is the last time step of the SBAS reconstruction.

Supp. Fig. 3: The following statement is made “most of the deformation at Coso takes place as transient deformation (which is what the model is sensitive to)” If the deformation was linear in time, wouldn't most of it still appear when summing up a series of outputs from the auto-encoder?

As shown in Fig. S5, the model is trained to be sensitive only to transient deformation. For it to be sensitive to steady deformation the model has to be retrained accordingly with different synthetic data.

Supp. Fig. 5 (and 3): It looks like there is an area of unwrapping errors. Is this the same area that is interpreted as an area of uplift (it looks like it is actually to slightly offset)? If so, it might be having a similar effect as introducing synthetic deformation as I suggested above. Some discussion of the effect of unwrapping errors would be useful, in any case.

We trained the model with synthetic incoherent pixels as well as unwrapping errors, and the model should not be sensitive to it. In Fig. formerly S5, now Fig. S11, the area with unwrapping errors is an area of low deformation in the outputs of the network. We have added a sentence that describes this in the main text: “Lastly, we add random pixels, in patches and isolated, to mimic incoherent pixels and unwrapping errors commonly encountered in real data.” (ll. 153-155)

Minor typo:

L22 's' missing from 'slow earthquake'.

Thanks for catching it!

Andy Hooper

Reviewer #2

InSAR has become one of the recent advances in high-resolution mapping of large-scale ground deformation on a regular basis. However, atmospheric phase delays have been a major factor that continues to hamper the accuracy of ground deformation measurements. Accurately extracting mm-scale deformation from InSAR timeseries hence is a critical yet challenging topic.

The authors propose a deep convolutional autoencoder to denoise InSAR timeseries that are contaminated by atmospheric noise, where the autoencoder learns the different spatial-temporal statistics between signal and noise, and outputs the cumulative displacement in the time series. To the best of my knowledge, this is the first study that utilizes inherent spatial-temporal statistics between signal and noise for denoising, though convolutional neural networks have been applied on individual interferograms in several previous studies. The proposed autoencoder is shown to be effective in extracting slow slip events on the NAF with 2-mm-level accuracy and ground deformation over the Coso geothermal field.

The study is interesting and promising. Once proved to be valid, the deep-learning based autoencoder could be a powerful tool to reveal ground deformation, especially for transient processes (e.g. SSE, transient creep) that had been previously hidden by atmospheric noises. I found the paper suitable for publication in Nature Communications once the authors addressed the comments below:

Thank you for your support, we have tried to address all your questions. Indeed, at times our main text was unclear on what is synthetic data and what is real data.

Questions about the autoencoder training:

1. At the 7th layer of the autoencoder, ground elevation is added. In the training set, is the topography randomly generated, or does it use the topography of the corresponding study area? Could you please elaborate on that. That is, can the pre-trained autoencoder be generalized to any study region if characteristics of DEMs are considered during training, or does the autoencoder need to be trained on a case-by-case basis?

The training DEMs are entirely synthetic and random (random roughness and power spectral density), such that the model can generalize to arbitrary DEM. We have added the following to the main text to make it clearer: "Spatially correlated Gaussian noise mimics delays from atmospheric turbulence of various length scales (Fig. 1 top row) and a quadratic function of pixels' elevation mimics the atmospheric delays that correlate with topography, also randomly generated (Jacobs 2017)".

2. If the input topography in the training set is from the study region, how is the topography is handled when converting it to the size of 40*40? Would sharp topographic boundaries be smeared out during the process? If so, the autoencoder may not be able to learn topography-correlated noise features below a certain spatial scale.

See answer above: the training DEMs are synthetic, and we do not have to rescale them to 48x48 (40 pixels was a typo, the size of the training synthetics is 48x48 pixels). Moreover, because the deep autoencoder is purely convolutional, the 48x48 input is not necessary to apply the model, which can be applied to arbitrary-sized (in space) time series.

3. For how many epochs did the training go? What is the training accuracy? Please provide the learning curve for the autoencoder (R^2 versus training epochs).

Thank you for pointing this omission! We have added as a new supplementary figure S2 the training curve as a function of number of training samples (the training data is infinite so there is no notion of epoch).

4. Is data normalization applied for input synthetic timeseries in the training set?

As the training data is purely synthetic, we generate it as already normalized (mean zero and standard deviation 1).

Questions about applications on real datasets:

5. The underlying assumptions for the implementation of the proposed autoencoder is that the signal is coherent whereas the atmospheric noises are random in time (Line 117). What if the deformation-signal duration is shorter than the time interval of the two SAR acquisitions? Would short transient signals hence be considered as noise?

For slow slip transient detection, this is probably not an issue for Sentinel and upcoming NISAR mission given their frequent recurrence interval. However, for ALOS-2, transient signals with <42 days duration may be falsely regarded as noise.

To better address this issue and evaluate the autoencoder performance, the authors could show the training accuracy versus the signal duration. A dependency may be expected. This evaluation is crucial when interpreting the cleaned timeseries outputs.

This is a very important point, indeed. A number of synthetic time series used in the training set actually have deformation taking place only between 2 successive acquisitions, and this has no impact on the performance of our model, as can be seen in the new figure S1 (last row, third column).

The input of our model is the cumulative range change and the output tentatively the actual cumulative deformation. Therefore, a rapid deformation taking place in-between acquisitions remains until the end of the time series and should be caught by the model.

6. In Figure 3, several sub-structures are observed along with the NAF slow slip events, e.g. features at (32.5°E, 48.88°N) and (32.63°E, 48.8°N). It is exciting to see those subtle features being extracted out of the noisy dataset. But it might also be residual noise. Any comments on those second-order features? What are the expectations when interpreting outputs from the autoencoder?

Reviewer 1 (Prof. Andy Hooper) had the same question, which we tried to address with the new supplementary figure S6 and the added text in the main manuscript: "What we presume to be remaining noise can be seen to the north-west and to the south-east of the slow earthquake in the output of the deep denoiser (see Fig. S7 in the Supplementary for cross-sections). We suspect these errors may arise from errors in the elevation model that propagated in the time series." (ll. 243-246)

7. The authors indicate that the current model interprets signals with wavelengths longer than 1 km as noise (Figure 4 caption). However, in the training set, the fault parameters are randomly set, including the fault depth. What I understand here is that the autoencoder is designed to not only identify surface shallow slip but also slip at depth.

What is the range of fault depth? The maximum wavelength it could resolve would correspond to the features of the input training set. Please clarify.

We realize this sentence was not clear. The wavelength limitation comes from the interplay between physical pixel size in the InSAR data and size of the training synthetics in pixels. In other words the wavelength limitation is really in number of pixels and depends on the size of the training synthetics. We have added the following text to the main manuscript to try to better explain this: “Our current model interprets wavelengths longer than a kilometer as noise, although experts might interpret those as the signature of slip at depth. This limitation however is related to the size of pixels with respect to the size of the training samples. The same network architecture trained on larger synthetics would circumvent this limitation (at the cost of increased computation and training time). An alternative approach consists in rescaling input data (see Supplementary figure S6) to ensure consistency of the model output in deformation wavelength (which is the case here for the NAF event).” (ll. 247-252).

We also added a new supplementary figure S6 to illustrate what we mean, with the following caption:

“The deep auto-encoder is trained on small synthetic time series (48x48 pixels), which limits the deformation wavelength it can resolve. As shown here, deformation beyond 20 to 30 pixels can be missed by the model, but down-sampling the input before applying the model enables to capture the full deformation. As in the previous figure, pre-existing deformation is not reconstructed by the model.”

8. Please indicate how the NSBAS network is formed in the Data Section, if it is not previously published. Figure S1 does not seem to match the description in Line 350. Missing figure?

Thank you for noticing this missing figure, we have put it back as figure S14 in the supplementary.

Reviewer #3

This paper introduces a deep auto-encoder that is designed to untangle ground deformation from atmospheric noise in InSAR time-series. I enjoyed reading the paper as it is very well written with clear and demonstrative examples.

Thank you for your comments! We have tried to answer all your questions and the details are below.

The following are some detailed comments:

(1) I find it slightly awkward branding de-noising InSAR cumulative deformation as "autonomous" extractions of deformation. InSAR processing itself does not involve prior knowledge of local tectonics. With good SNR, even the simplest InSAR processing can manifest deformation that can be identified without expert eyes -- and we do not brand SBAS as a method to "autonomously" extract signals. The presented model is an additional step in standard InSAR processing (since it requires the input of InSAR time-series) designed to retrieve small signals in low SNR scenarios. The extracted deformation from the presented model still needs expert interpretation to avoid false positives, whether because the synthetic noise model in the training sets does not represent actual noise in the data or because the model failed with very low SNR.

Our argument for using the 'autonomous term' is twofold: (i) our model explicitly separates noise from different sources of deformation, (ii) it does not rely on a priori knowledge of the physical source of deformation for identifying such signal. For instance, in the case of the NAF slow slip event, we do not provide, at any time, any information about the fault surface trace or position at depth. In Fig. 3 of the main text we show the fault trace for visual reference, but the model does not have access to it. In this sense, we believe our model to be autonomous. We agree 'autonomous' may be a strong word but 'automatic' on the other hand would be too weak in our opinion.

(2) Given how much the performance of the model heavily depends on the design of the relative spatial wavelengths of the noise and signal in the training sets (the authors noted, first in Fig. 4 that the model interprets wavelength longer than 1 km as noise and then in the discussion that the input pixel size of the InSAR data restricts the size of the deformation signal that can be deciphered), I am interested to see how well temporal filtering worked. Given that making use of different temporal signatures of deformation and atmospheric signals is the main novel aspect in terms of the model design of the paper, I think the paper will benefit from some discussions on this aspect.

This is a very good point, and we have made tests on a temporal filter that consists in comparing the temporal average after the fault slip with the temporal average before the fault slip. This is indeed an important comparison to make, and we have added a figure to the supplementary (S3), which we refer to in the main text: "The performance of a simple temporal filter is shown for comparison in the Supplementary (Fig. S3)." (l.193).

Reviewers' Comments:

Reviewer #1:

Remarks to the Author:

Thank you for addressing all my comments. I reiterate that, in my opinion, this paper is an excellent contribution, and I would be very happy to see it published.

Andy Hooper

Reviewer #2:

Remarks to the Author:

The authors have addressed my comments and I really appreciate the synthetic tests that added to the supplementary materials. I only have several minor comments below:

Figure 1: the size of training synthetics should be updated to 48x48?

L89: 'validation'

Figure 2, L177-L179: test sample size not consistent in text and figure caption.

L179-L180: the definition of SNR should be updated according to Eq 6, 7.

L340: the calculation of trainable parameters in each layer may be incorrect. Please double check.

Reviewer #3:

Remarks to the Author:

Thank you for adding the comparison of a simple temporal filter (stacking). The advantage of the presented method over a simple stacking approach is clear. However, the comparison is not fair since the presented method also has spatial filtering. What I intended to ask was the performance of auto-encoding ONE interferogram of cumulative deformation with noise (the last scene of the input InSAR time series, rather than a sequence of InSAR time-series) so that there is spatial filtering but no temporal filtering is involved. I am curious how much spatial filtering contributed to de-noising comparing to that of temporal filtering. Apologies if my previous comments were not clear.

Response to Reviewers

We are very grateful once again for the time and effort spent by the reviewers on our paper, and we have tried to address every comment in the detailed response below.

In what follows the reviewers' comments are in black and our answers are in blue. The edited text in the manuscript and its supplementary is also in blue.

In addition to addressing reviewer 3's suggestion of creating and training a new single-timestep neural network to compare with the 9-steps network analyzed so far, we have added a few modifications to further address the previous round of reviews.

Reviewer #1 (Remarks to the Author):

Thank you for addressing all my comments. I reiterate that, in my opinion, this paper is an excellent contribution, and I would be very happy to see it published.

Andy Hooper

Dear Prof. Hooper, many thanks for your support and your warm encouragements!

Reviewer #2 (Remarks to the Author):

The authors have addressed my comments and I really appreciate the synthetic tests that added to the supplementary materials.

Thank you!

I only have several minor comments below:

Figure 1: the size of training synthetics should be updated to 48x48?

Absolutely, thank you for catching this.

L89: 'validation'

Figure 2, L177-L179: test sample size not consistent in text and figure caption.

L179-L180: the definition of SNR should be updated according to Eq 6, 7.

Indeed, thank you for noticing this, we had not updated this section during the first round of reviews, hence the discrepancy.

We also updated the figures and Methods with the classic definition of RMS suggested by Reviewer 1, squared compared to the previous definition: $(\text{RMS of the signal})^2 / (\text{RMS of the noise})^2$.

L340: the calculation of trainable parameters in each layer may be incorrect. Please double check.

The number of trainable parameters was correct, but we clarified the number of parameters for each layer in the network (ll. 340-348).

Reviewer #3 (Remarks to the Author):

Thank you for adding the comparison of a simple temporal filter (stacking). The advantage of the presented method over a simple stacking approach is clear. However, the comparison is not fair since the presented method also has spatial filtering. What I intended to ask was the performance of auto-encoding ONE interferogram of cumulative deformation with noise (the last scene of the input InSAR time series, rather than a sequence of InSAR time-series) so that there is spatial filtering but no temporal filtering is involved. I am curious how much spatial filtering contributed to de-noising comparing to that of temporal filtering. Apologies if my previous comments were not clear.

We now also show the comparison with a neural network that has the same architecture, but only has a single timestep as input, so that there is now also a comparison with a pure spatial filtering. We did so by training from scratch a model that has the same architecture but a single time step as input.

We added the following discussion of these new results in the main text: “For comparison, the performance on synthetic data of the same architecture shown in Fig. 1 but trained on single time steps is shown in Fig. S3.” (ll. 191-192).

These results are shown in the new figure 3 of the supplementary, with the following caption: “To evaluate the effect of the temporal filters learned by our model, we trained from scratch another model, trained to detect deformation in a single time step instead of 9. Everything is kept the same, as described in the main text. Top: Median performance (10^5 test samples) of the single step model (in red) and the 9-steps model analyzed in the main text and the rest of the supplementary (in black), as measured by structural similarity index (SSIM) between models output and deformation ground truth, as a function of signal to noise ratio (SNR). Bottom: examples of the data showing input, ground truth, its reconstruction by the deep autoencoders, 9-steps on the left and single step on the right. For a given threshold of 0.4 SSIM, the single time step model fails at 2% SNR and the initial 9-steps models fails at 0.6%, about a third the amount of noise. Note moreover that this single-step model would fail the tests shown in Fig. S6 below”

Reviewers' Comments:

Reviewer #3:

Remarks to the Author:

I appreciate the authors' efforts in adding a performance comparison of a pure spatial filter. I am happy to see this paper published as is.